# Heavy Metal Ions Trigger a Fluorescent Quenching in DNA–Organic Semiconductor Hybrid Assemblies

**DOI:** 10.3390/polym14173591

**Published:** 2022-08-31

**Authors:** Xianyang Li, Yuhui Feng, Tao Yi, Yan Piao, Dong Hyuk Park, Longzhen Cui, Chunzhi Cui

**Affiliations:** 1National Demonstration Centre for Experimental Chemistry Education, Department of Chemistry, Yanbian University, Yanji 133002, China; 2Jilin Tobacco Industry Co., Ltd., Yanji 133001, China; 3Longjing Municipal Ecological and Environmental Monitoring Station, Longjing 133400, China; 4Department of Chemical Engineering, Inha University, Incheon 22212, Korea; 5Department of Preventive Medicine, Yanbian University, Yanji 133002, China

**Keywords:** DNA, heavy metal ions, tris-(8-hydroxyquinoline)aluminum

## Abstract

The significance of DNA is no longer limited to its role as a biological information carrier; as a natural polymer, it also become in the field of materials. Single-stranded DNA (ssDNA) molecules with specific sequences can form a G-quadruplex or hairpin-shaped conformation with specific heavy metal ions through coordination bonds. In this study, ssDNA molecules of the four sequences were prepared into hybrid assemblies with one of the famous display materials, the tris-(8-hydroxyquinoline)aluminum (Alq_3_) semiconductor. Based on these hybrid assemblies, heavy metal ions, namely Pb^2+^, Hg^2+^, Cd^2+^ and As^3+^, were detected individually at the ppb level. Apart from this, in practical application, many samples containing heavy metal ions are digested with acid. By introducing MES buffer solution, the influence of acidity on the fluorescent signal of Alq_3_ was excluded. This strategy showed promising results in the practical application of detecting heavy metal ions in shrub branches and leaves.

## 1. Introduction

With industrialization, urbanization and the frequent use of chemical fertilizers, the situation of soil pollution by heavy metals has become serious. Through food chain accumulation, plants or crops contaminated with heavy metals could have negative effects on human health. The ionic form of these heavy metals can destroy the endocrine and nervous systems, causing motion and cognitive disorders, kidney failure and even DNA damage [1,2]. Hence, quick and sensitive detection methods are of considerable importance for environmental protection and human health. In order to make up for various shortcomings of conventional analytical techniques, such as requiring expensive equipment, time-consuming procedures and highly-trained technicians, numerous methods have been developed to detect these metal ions, including electrochemical techniques [3,4,5,6,7,8,9,10,11,12], spectroscopy [13,14,15,16,17,18] and colorimetry [19,20,21,22,23,24]. Among these, aptamer-based assays for detection of the heavy metal ions have received significant attention owing to their appropriate selectivity, great sensitivity, low cost, facile operation and wide response range [25,26,27,28,29,30]. In general, sequence-designed aptamers and heavy metal ions can form a G-quadruplex or hairpin conformation, and this provides the premise for detection [31,32]. In the previous research of our group, we developed hybrid assemblies consisting of aptamers and organic semiconducting small molecules and applied them to the detection of mercury ions [33,34]. This detection approach was based on a significant “on–off” phenomenon of the fluorescent hybrid assemblies, and their selectivity was also evaluated using other metal ions, such as potassium, sodium, zinc, chromium, cadmium and copper. However, the approach was found to have a fatal shortcoming, a lack of acid resistance which induces the fluorescent quenching of these organic molecules in an acidic environment. Typically, actual samples in which heavy metal ions need to be detected are processed by acid digestion. Therefore, in the detection of heavy metal ions based on organic fluorescent probes, solving the lack of acid resistance is an unavoidable task. 

In this research, four single-stranded DNA (ssDNA) sequences corresponding to Pb^2+^, Hg^2+^, Cd^2+^ and As^3+^ were designed and prepared in hybrid assemblies together with an organic semiconducting molecule, tris-(8-hydroxyquinoline)aluminum (Alq_3_). In order to solve the problem of acid resistance, we optimized the detection approach and finally achieved acid resistance using MES buffer solution. After that, multiplex detection of the above four heavy metal ions was completed. As a practical application, this optimized approach was also used for the detection of heavy metal ions in shrub branches and leaves. These results will broaden the detection approaches for heavy metal ions based on DNA molecules and provide a meaningful solution to the weak acid resistance of organic fluorescent probes in actual application.

## 2. Materials and Methods

### 2.1. Materials

Tris-(8-hydroxyquinoline)aluminum (Alq_3_, 99.995%), 2-morpholinoethanesulfonic acid (MES, 99%) and 2-[4-(2-hydroxyethyl)piperazin-1-yl]ethanesulfonic acid (HEPES, 99%) were purchased from Sigma-Aldrich. Tetrahydrofuran (THF, 99%) was purchased from Aladdin. ssDNA and Tris-HCl buffer solution were purchased from Shanghai Shenggong Biological, China. Sequences of ssDNA which can specifically bind with four heavy metal ions are as follows: Pb^2+^: 5′-GGTTGGTGTGGTTGG-3′;Hg^2+^: 5′-TTCTTTCTTCCCCTTGTTTGTT-3′;Cd^2+^: 5′-GGGTTCACAGTCCGTT-3′;As^3+^: 5′-ATGCAAACCCTTAAGAAAGTGGTCGTCCAAAAAACCATTG-3′.

Nitric acid (HNO_3_, 65.0~68.0%) was purchased from Sinopharm Chemical Reagent. Shrub branches and leaves were obtained from the Institute of Geophysical and Geophysical Exploration, Chinese Academy of Geological Sciences.

### 2.2. Preparation of Hybrid Assemblies

Commercially available Alq_3_ powders were dissolved in THF at a concentration of 1 mg/mL to form a stock solution. On the other hand, ssDNA was dissolved in deionized water (DI water) at a concentration of 500 nM. Then, the stock solution (2 mL) was injected into 20 mL of ssDNA solution under vigorous stirring (800 rpm) for 2 min. Subsequently, the mixture was kept at room temperature for 10 h to allow the formation of a visible precipitate.

### 2.3. Reaction with Heavy Metal Ions

One milliliter of hybrid assembly solution was mixed with various concentrations (50–400 ppb) of heavy metal ions at a volume ratio of 1:1 and kept for at least 30 min.

### 2.4. Exploration of the Conditions for the Detection of Heavy Metal Ions

One milliliter of hybrid assembly solution was thoroughly mixed with buffer solution (HEPES, Tris-HCl and MES) at a volume ratio of 1:1 and then mixed with 2 mL of Pb^2+^ (100–400 ppb) with various concentrations and incubated for 30 min.

### 2.5. Reaction with HNO_3_

One milliliter of hybrid assembly solution was mixed with various concentrations (0.03–3%) of HNO_3_ in a volume ratio of 1:1 and incubated for 30 min.

The hybrid assembly solution was thoroughly mixed with MES buffer solution at a volume ratio of 1:1 and then mixed with various concentrations (0.5–4%) of HNO_3_ at a volume ratio of 1:1 and incubated for 30 min.

### 2.6. Detection of Heavy Metal Ions in Actual Samples

In the preparation of actual sample, 0.5 g of shrub branches and leaves were weighed and placed in a microwave digestion tank. Then, 5 mL of HNO_3_, 1 mL of H_2_O_2_, 1 mL of HCl and 1 mL of HF were poured into the tank, which was then sealed and placed in a microwave digestion apparatus (Multiwave3000, Anton Paar, Graz, Austria). Digestion was performed at room temperature, 120 °C and 160 °C, for 5 min at each temperature, and then at 200 °C for 30 min. When the temperature dropped below 40 °C, the digestion tank was taken out, and the mixture solution was dried to 1 mL under heating at 130 °C. After cooling to room temperature, the solution was transferred to a volumetric flask and made up to volume using DI water.

For the detection of heavy metal ions, before each test of the actual sample, the standard curve was redrawn. Two milliliters of hybrid assembly solution was thoroughly mixed with MES buffer and then mixed with actual sample solution at a volume ratio of 1:1 and incubated for 30 min. Then, the mixtures were characterized using a fluorescence spectrophotometer with an excitation wavelength of 370 nm.

### 2.7. Characterization

A scanning electron microscope (Hitachi, SU8010, Tokyo, Japan) was used to observe the morphology of the sample, and a fluorescence spectrophotometer (Jasco, FP-8200, Tokyo, Japan) was used to observe the photoluminescence spectra of the samples.

## 3. Results

Figure 1 shows the scanning electron microscopy (SEM) images of the hybrid assemblies prepared by a facile re-precipitation method [35]. The ssDNA sequence was 5′-GGTTGGTGTGGTTGG-3′, which could specifically bind with Pb^2+^. The as-prepared hybrid assemblies showed a hexagonal prism shape with a smooth surface, as shown in Figure 1a. After exposure to the Pb^2+^ ion solution, cracks began to appear on the smooth surface of the hybrid assemblies. As the concentration of the Pb^2+^ ion solution increased, the cracks became more obvious.

Meanwhile, the photoluminescence (PL) spectra of the hybrid assemblies before and after the interaction with the Pb^2+^ ion solution were observed, as shown in Figure 2a. At first, a broad PL peak of hybrid assemblies was observed at ~510 nm. The intensity of the PL peak was observed to decrease drastically upon the interaction with different concentrations of the Pb^2+^ ion solution. Phenomena such as the appearance of cracks and the decrease in PL peak intensity were also observed in previous studies. The reason for the PL quenching is attributed to electron transfer from Alq3 molecules to the mercury ions due to the bridge role of the hairpin conformation of ssDNA [33]. Based on these phenomena, the detection of Pb^2+^ was realized. It is worth noting that these heavy metal ions were prepared in DI water. However, in practical application, we need to consider the effect of an acidic environment on the PL signal of hybrid assemblies, since many samples are digested by acid. Figure 2b shows the influence of the acidic environment on the PL signal of hybrid assemblies. Obviously, when these hybrid assemblies were exposed to an acidic environment, the PL intensity decreased with increasing acid concentration.

Therefore, in order to exclude the interference of acidity in the detection of heavy metal ions, strategies for using buffer solutions have been introduced. Figure 3a–c show the PL spectra of the hybrid assemblies treated with HEPES, Tris-HCl and MES buffer solution, respectively, and then exposed to various concentrations of Pb^2+^ ion solutions. Interestingly, the cases of treatment with HEPES and Tris-HCl maintained the original PL intensities after exposure to various concentrations of Pb^2+^ ion solutions. This may be due to the fact that Pb^2+^ ions first form complexes with the anionic groups in HEPES or Tris-HCl buffer solution, thus failing to specifically bind to ssDNA molecules of the hybrid assemblies. The PL intensity of the hybrid assemblies decreased only in the case of MES buffer solution. In addition, an acid resistance test of the hybrid assemblies treated with MES buffer was performed, as shown in Figure 3d. It can be seen that the hybrid assemblies treated with MES buffer exhibited resistance to an acidic environment. Hence, we can infer that the PL quenching of the hybrid assemblies shown in Figure 3c is caused by the recognition of lead ions.

Similarly, the other three heavy metal ions, Hg^2+^, Cd^2+^ and As^3+^, were detected by using the hybrid assemblies containing the corresponding ssDNA sequences. As the concentration of heavy metal ions increases, the PL intensities of the hybrid assemblies also decrease sharply, as shown in Figure 4.

Under the optimized conditions, the relationship between the PL intensities of different hybrid assemblies and the concentration of corresponding target ions was investigated. The PL intensities showed a linear relationship with the level of the heavy metal ions in the range from 50 ppb to 350 ppb, as shown in Figure 5. Correlation coefficients (R^2^) are 0.9945, 0.991, 0.9957 and 0.9917 for Pb^2+^, Hg^2+^, Cd^2+^ and As^3+^, respectively. The limit of quantification (LOQ) is defined as 10S_0_/S, where S_0_ is the standard deviation and S is the slope of the calibration curve [36]. According to this formula, the LOQs of Pb^2+^, Hg^2+^, Cd^2+^ and As^3+^ were calculated to be 22.92 ppb, 21.74 ppb, 12.68 ppb and 49.29 ppb, respectively.

In order to verify the feasibility of the detection method in actual samples, shrub branches and leaves were selected. Their heavy metal ion content was determined using inductively coupled plasma mass spectrometry (ICP-MS). Figure 6 shows the estimated results for the content of heavy metal ions in the sample: 72.21 ppb, 2.00 ppb and 10.27 ppb for Pb^2+^, Cd^2+^ and As^3+^, respectively. The results were all within the reference concentration range. The result that Hg^2+^ was not detected in the sample was also consistent with the results provided by ICP-MS.

## 4. Conclusions

In this study, we developed a heavy metal ion probe based on hybrid assemblies of ssDNA molecules and organic semiconducting molecules, and we achieved low-concentration detection of four heavy metal ions, namely Pb^2+^, Hg^2+^, Cd^2+^ and As^3+^. In addition, by introducing MES buffer solution, the problem of the detection signal of these probes being affected in an acidic environment was solved. Based on this, the practical application purpose of evaluating the content of heavy metal ions in shrub branches and leaves was also achieved. These findings provide meaningful guidance for detection probes based on ssDNA or aptamers, especially for the interference of fluorescent probes in acidic environments.

## Figures and Tables

**Figure 1 polymers-14-03591-f001:**
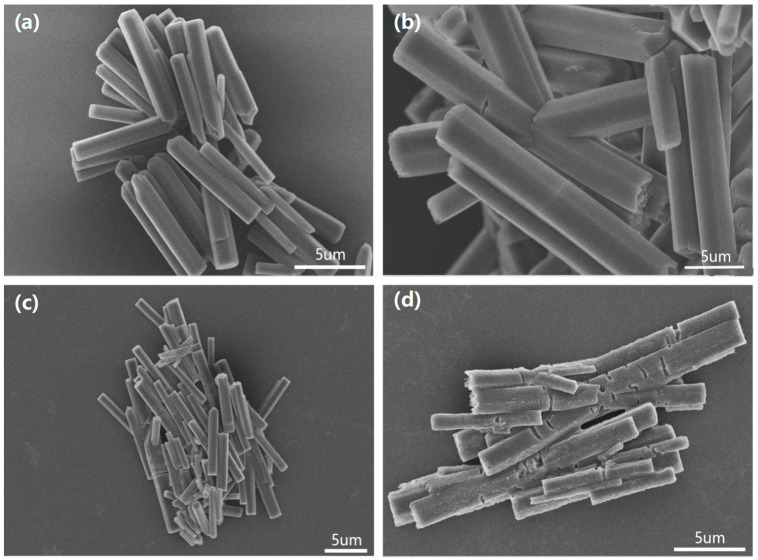
SEM images of (**a**) as-prepared hybrid assemblies and hybrid assemblies exposed to (**b**) 0.1 ug/mL, (**c**) 1 ug/mL and (**d**) 10 ug/mL of Pb^2+^ solution.

**Figure 2 polymers-14-03591-f002:**
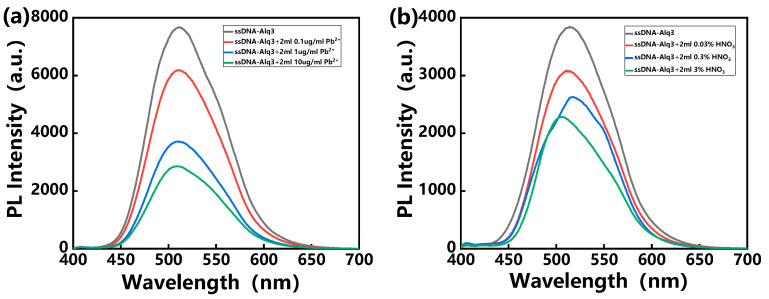
PL spectra of hybrid assemblies exposed to various concentrations of (**a**) Pb^2+^ solution and (**b**) HNO_3_ solution, with excitation at 370 nm.

**Figure 3 polymers-14-03591-f003:**
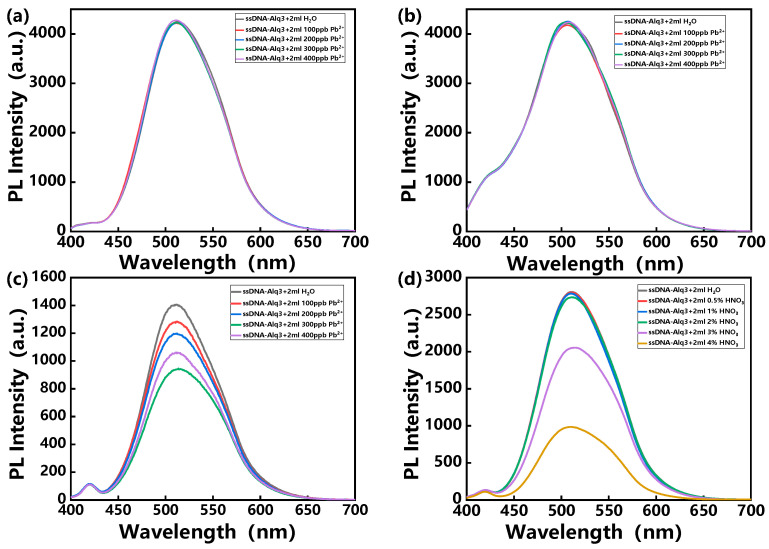
PL spectra of hybrid assemblies treated with (**a**) HEPES, (**b**) Tris-HCl and (**c**) MES buffer solution and exposed to various concentrations of Pb^2+^ solution, with excitation at 370 nm. (**d**) PL spectra of hybrid assemblies treated with MES buffer solution treated and exposed to HNO_3_ solution, with excitation at 370 nm.

**Figure 4 polymers-14-03591-f004:**
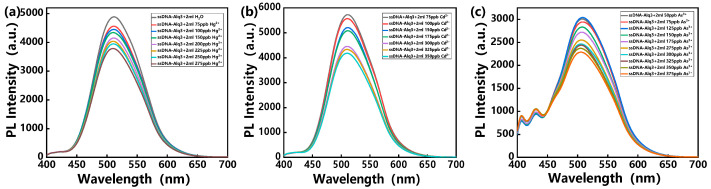
PL spectra of hybrid assemblies treated with MES buffer solution and exposed to various concentrations of (**a**) Hg^2+^, (**b**) Cd^2+^ and (**c**) As^2+^ solution, with excitation at 370 nm.

**Figure 5 polymers-14-03591-f005:**
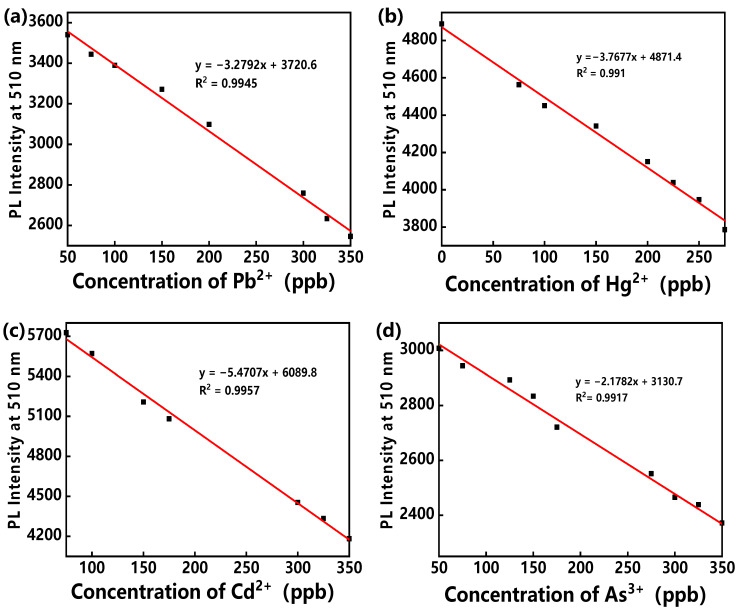
The relationships between PL intensities of MES-treated hybrid assemblies and various concentrations of (**a**) Pb^2+^, (**b**) Hg^2+^, (**c**) Cd^2+^ and (**d**) As^3+^.

**Figure 6 polymers-14-03591-f006:**
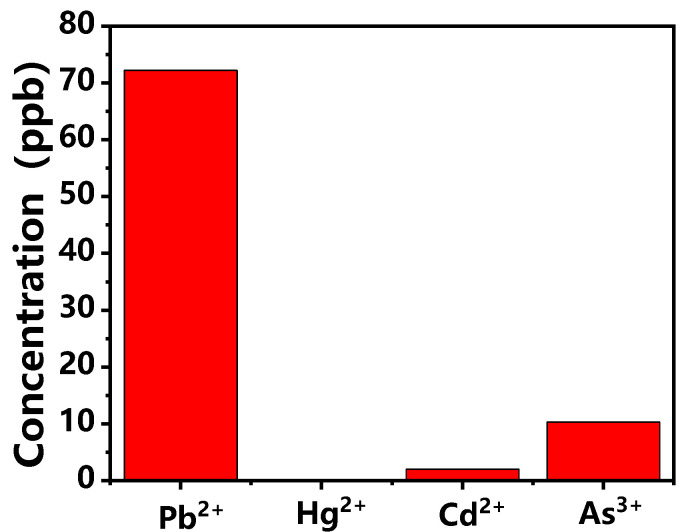
Evaluation results for four heavy metal ions in shrub branches and leaves.

## Data Availability

Not applicable.

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
