# Peer review of "Heavy Metal Ions Trigger a Fluorescent Quenching in DNA–Organic Semiconductor Hybrid Assemblies"

_polymers, 2022, doi:10.3390/polym14173591_

Round 1

Reviewer 1 Report

In the present manuscript, Feng, Piao, Cui and co-workers discuss the synthesis and characterization of ssDNA -tris (8-hydroxyquinoline) aluminum (Alq3) hybrid. The luminescent properties of the DNA-metal complex have been utilized for sensing heavy metal ions. The sensitivity of the sensor system depends on the particular sequence of ssDNA that eventually binds with various heavy atom metal ions and consequent quenching in photoluminescence of the DAN-Alq3 hybrid. Furthermore, the practical application of the sensor system is demonstrated by conducting studies in acidic conditions as heavy metals are often digested by acid. The critical part of this study is the identification of a suitable buffer for retaining the sensing property of a hybrid system even in acidic conditions. I think the work is interesting and suitable for publication in polymers once the authors address the following minor aspects mentioned below.

Specific points:

  1. Abstract: “……hairpin-shaped conformation with corresponding heavy metal ions…..” the word ‘corresponding’ makes the sentence unclear
  2. Materials and Methods: (i) ‘ssDNA molecules’ it would be better to use only ‘ssDNA’ (ii) Preparation of hybrid assemblies: ssDNA solution – the detail of solvent used is missing. (iii) The purpose of reaction with heavy metal ions is unclear. (iv) ….of HNO3 in as a volume ratio of 1:1…” remove the term ‘as.’
  3. Results: “……hybrid assemblies prepared by a facile re-precipitation method [35].” It would be helpful for readers if the authors provided the details of the re-precipitation method used to prepare the hybrid assemblies in the Materials and Methods section. 
  4. A reason for the quenching fluorescence ssDNA-Alq3 hybrid in the presence of heavy metal ions may include.
  5. Figure 2-4. The excitation wavelength (lex) used for the measurements is missing.
  6. Provide a suitable reference for the equation used for the deter4mination of LOD.
  7. Figure 5. The Y-axis legend should be the wavelength photoluminescence monitored instead of PL intensity (a. u.)
  8. The authors may include the details of photoluminescence sensing studies conducted in different conditions in the Materials and Methods section.
  9. It would be better if the authors briefly discussed the reason behind the selection of shrub branches and leaves for the present study.

Author Response

We thank very much to Reviewers. Their comments help to improve the quality of the revised manuscript. We have answered all of the comments raised by the Reviewers.

Reviewer (#1)’s comments:

In the present manuscript, Feng, Piao, Cui and co-workers discuss the synthesis and characterization of ssDNA -tris (8-hydroxyquinoline) aluminum (Alq3) hybrid. The luminescent properties of the DNA-metal complex have been utilized for sensing heavy metal ions. The sensitivity of the sensor system depends on the particular sequence of ssDNA that eventually binds with various heavy atom metal ions and consequent quenching in photoluminescence of the DAN-Alq3 hybrid. Furthermore, the practical application of the sensor system is demonstrated by conducting studies in acidic conditions as heavy metals are often digested by acid. The critical part of this study is the identification of a suitable buffer for retaining the sensing property of a hybrid system even in acidic conditions. I think the work is interesting and suitable for publication in polymers once the authors address the following minor aspects mentioned below.

Specific points:

(1) Abstract: “……hairpin-shaped conformation with corresponding heavy metal ions…..” the word ‘corresponding’ makes the sentence unclear

Response:

  We have replaced the word “corresponding” with “specific” in the revised version, page 1, line 16.

(2) Materials and Methods: (i) ‘ssDNA molecules’ it would be better to use only ‘ssDNA’ (ii) Preparation of hybrid assemblies: ssDNA solution – the detail of solvent used is missing. (iii) The purpose of reaction with heavy metal ions is unclear. (iv) ….of HNO3 in as a volume ratio of 1:1…” remove the term ‘as.’

Response:

(i) We have replace “ssDNA molecules” with “ssDNA” in the revised version, page 2, line 67.

(ii) We have indicated the solvent (DI-water) used to prapare ssDNA solution in the revised version, in the part of rewritten “Preparation of hybrid assemblies”, page 2, line 82-86.

(iii) I'm so sorry I'm not quite sure what exactly you mean by the purpose of reacting with heavy metal ions. In the introduction part, we have explained the purpose and significance of detecting heavy metal ions. If you think it is not enough, we will increase the content of this part.

(iv) We have removed the the term “as”, in the revised version, page 3, line 97.

(3) Results: “……hybrid assemblies prepared by a facile re-precipitation method [35].” It would be helpful for readers if the authors provided the details of the re-precipitation method used to prepare the hybrid assemblies in the Materials and Methods section.

Response:

Thank you for your suggestion. We have rewritten the part of  “Preparation of hybrid assemblies”. The details of the re-precipitation method are included, in the revised version, page 2, line 82-86.

(4) A reason for the quenching fluorescence ssDNA-Alq3 hybrid in the presence of heavy metal ions may include.

Response:

Thank you for your suggestion. We have added the queching mechamism in the revised version, page 3, line 136-138.

(5) Figure 2-4. The excitation wavelength (λex) used for the measurements is missing.

Response:

Thank you for your suggestion. We have indicated the excition waveleng usde for the measurements in the caption of Figure 2-4, and in the Materials and Methods section.

(6) Provide a suitable reference for the equation used for the determination of LOD.

Response:

Thank you for your suggestion. First of all, we apologize for the flagging error. Actually what we calculated is “limit of quantification (LOD)”, not “limit of detection (LOD)”. We have revised it and provide a suitable reference for the equation in the revised version, page 5, line 182-185, and new reference 36.

(7) Figure 5. The Y-axis legend should be the wavelength photoluminescence monitored instead of PL intensity (a. u.)

Response:

Thank you for your suggestion. We have replaced the previous “Figure 5” with the revised one in the revised version, page 6, line 187-188.

(8) The authors may include the details of photoluminescence sensing studies conducted in different conditions in the Materials and Methods section.

Response:

Thank you for your suggestion. We have added the details of photoluminescence sensing studies conducted in different conditions in the Materials and Methods section, page 3, line 95-118.

(9) It would be better if the authors briefly discussed the reason behind the selection of shrub branches and leaves for the present study.

Response:

Thank you for your suggestion. We have briefly discussed the reason behind the selection of shrub branches and leaves in the revised version, page 192-195.

Reviewer 2 Report

This study reports on the test and evaluation of a new method for heavy metal ion probes based on hybrid assemblies of ssDNA molecules and organic semiconducting molecules. It is claimed that the novel method can detect low levels (PPB) of heavy metal ions, including Pb2+, Hg2+, Cd2+, and As3+. The results are convincing, and the conclusions are supported by the data. Furthermore, the paper is clear and well-organized. I do recommend this work for publication in Polymers.

As I minor recommendation, I suggest improving the description of the results using real leaves from plants. Reading through, the manuscript I did not understand how the samples prepared using real plant leaves are processed. The processing of these samples should be described in more detail.

Author Response

We thank very much to Reviewers. Their comments help to improve the quality of the revised manuscript. We have answered all of the comments raised by the Reviewers.

Reviewer (#2)’s comments:

This study reports on the test and evaluation of a new method for heavy metal ion probes based on hybrid assemblies of ssDNA molecules and organic semiconducting molecules. It is claimed that the novel method can detect low levels (PPB) of heavy metal ions, including Pb2+, Hg2+, Cd2+, and As3+. The results are convincing, and the conclusions are supported by the data. Furthermore, the paper is clear and well-organized. I do recommend this work for publication in Polymers.

As I minor recommendation, I suggest improving the description of the results using real leaves from plants. Reading through, the manuscript I did not understand how the samples prepared using real plant leaves are processed. The processing of these samples should be described in more detail.

Response:

Thank you for your suggestion. We have added the details of preparation of actual samples in the Materials and Methods section, page 3, line 102-118.